# Evaluation of the Conservation Status of the Croatian Posavina Horse Breed Based on Pedigree and Microsatellite Data

**DOI:** 10.3390/ani11072130

**Published:** 2021-07-18

**Authors:** Ante Ivanković, Giovanni Bittante, Miljenko Konjačić, Nikolina Kelava Ugarković, Mateja Pećina, Jelena Ramljak

**Affiliations:** 1Faculty of Agriculture, University of Zagreb, Svetošimunska 25, 10000 Zagreb, Croatia; mkonjacic@agr.hr (M.K.); nkelava@agr.hr (N.K.U.); matejapecina@agr.hr (M.P.); jramljak@agr.hr (J.R.); 2Department of Agronomy, Food, Natural Resources, Animals and Environment (DAFNAE), University of Padova, Via dell’Università 16, 35020 Legnaro, Italy; bittante@unipd.it

**Keywords:** Croatian Posavina horse, pedigree, phenotype, genetic diversity, conservation

## Abstract

**Simple Summary:**

Conservation of local horse breeds as part of animal genetic resources is of national and global importance. Monitoring of local breeds is often fragmentary, i.e., it involves analysis of pedigrees, phenotype, and genetic structure. Using the Croatian Posavina horse as an example, we analyzed the status of the population with regard to available pedigree information, phenotype measures of stallions and mares, and genetic structure based on microsatellites. The generation interval is about eight years, indicating relatively early involvement of animals in reproduction for economic use of the breed. The depth of the pedigree is relatively modest due to a relatively short period of systematic breeding work (two decades). The number of active sire-lines and mare-lines is favorable and forms a good basis for the preservation of the breed. Regarding conformation, the Croatian Posavina horse kept the recognizability of the small-sized horse breed in the coldblooded type, in which there are clear traces of the earlier controlled introduction of the Arabian and other breeds. Its genetic diversity component has been preserved. The above results are a guide for further implementation of effective programs for the conservation of endangered local horse breeds.

**Abstract:**

The Croatian Posavina horse (CPH) is native Croatian breed under a conservation program and under various programs of economic use (ecosystem services, agrotourism, and meat production). The aim of this study was to analyze the status of the CPH population through an analysis of their pedigree (28,483 records), phenotype (292 licensed stallions, 255 mares), and genetic structure (292 licensed stallions). The average generation interval was 8.20 years, and the number of complete generations was 1.66. The effective number of founders and ancestors was 138 and 107, respectively, with a ratio of 1.29, and the genetic conservation index was 4.46. As for the morphometric characteristics, the average withers height of the stallions was 142.79 cm, the chest circumference was 194.28 cm, and the cannon bone circumference was 22.34. In mares, the withers height, chest, and cannon bone circumference were lower (139.71 cm, 190.30 cm, and 20.94 cm, respectively). Genetic microsatellite analysis of the 29 sire-lines showed high genetic diversity, expressed as the mean allele number (7.7), allele richness (4.0), and expected heterozygosity (0.740). There was no evidence of high inbreeding or a genetic bottleneck. The genetic and phenotypic data indicate that the CPH is an important and diverse reservoir of genetic diversity and can be conserved because of its special characteristics (adaptability).

## 1. Introduction

Horse breeds are undoubtedly an important part of the mosaic of national, regional, and global genetic resources. In the 19th and 20th centuries, local horse breeds, especially of the coldblooded type, were subjected to extinction pressure due to the loss of their function in agriculture. During the 20th century, three local horse breeds, which represented the largest group of ponies (Krk Island pony, Dalmatian bušak, and Lika horse), were lost in Croatia. Of the total number of breeds of mammalian species used for agriculture in the world, horse breeds account for 14.96% [1]. Although the number of extinct horse breeds is relatively low (9.8%), a large number of breeds are at risk (33.4%), and for most horse breeds their endangered status is unknown (54.4%) [1]. Coldblooded horse breeds are primarily threatened by the loss of their previous working usefulness in agriculture, while warmblood horse breeds are generally in a better position. However, today’s public is aware of the need to preserve local breeds for a number of secondary benefits, thanks to global and national activities (Convention on Biological Diversity, The Global Plan of Action for Animal Genetic Resources, National Plan on the Conservation of Animal Genetic Resources, and other documents). Local breeds are part of the identity and tradition of the region, maintaining rural vitality and local gastronomy, adaptability, ecosystem services, and landscape conservation.

One such breed is the Croatian Posavina Horse (CPH), whose primary genetic base was the archaic horse of areas of the Sava River Basin (see Appendix A). For centuries, the breeding area of the CPH was at the intersection of the Habsburg and Ottoman Empires, and this geopolitical position of the breeding area left its mark on the conformation, adaptability, and usability of the CPH. At a time when the need for a horse suitable for agriculture and war use was often changing, a smaller, adaptable, and working/riding horse with simultaneously restrictive feeding needs emerged. From the 17th to the 20th century Arabian stallions that were purchased or captured in war activity were occasionally used to genetically improve the stamina and speed of the CPH. Until the second half of the 20th century, the CHP has retained its recognition value, which was partially compromised by the limited introduction of several coldblooded horse breeds in the second half of the 20th century, primarily to increase muscularity and carcass composition. Like all other coldblooded horse breeds, the CPH suffered a population decline in the 20th centuries following the onset of the industrialization process in agriculture [2,3,4]. In the nineties of the 20th century, the CPH population declined to about 450 breeding individuals. Active measures to support the preservation of the breed (e.g., defined breeding program, breeding organization activities, annual subsidies, permanent promotion, and efficiency in meat production) helped to revive its breeding. Currently, the value of the CPH is reflected through horse meat as the main economic income, activities at community levels, and as a genetic resource. The breed has an ecoservice function (maintaining the seasonally flooded pastures of the Sava River Basin and protected natural parks) through traditional grazing systems. The actual size of the CPH population is 5854 individuals (300 stallions, 2877 mares, and 2677 offspring) and falls into the “vulnerable” category [5].

Preserving breeds is a prerequisite for maintaining genetic diversity and relationships within and between populations [4]. If the relationships between different populations are ignored, remarkable genetic erosion can occur on the global scale [6] and ultimately result in the risk of losing a large portion of the equine genome. The genealogical data are the basis of selection work, especially in the monitoring of conservation strategy and genetic variability of vulnerable horse breeds. The usefulness of a pedigree analysis has been observed in numerous studies, e.g., the Lipizzan horse [7], Andalusian horse [8], Asturcón pony [9], Noriker [10], Slovak horse breeds [11], Sorraia [12], Turkish Arab horse [13], Hucul horse [14], Italian Maremmano horse breed [15], and others. Pedigree analyses allow to assess the inbreeding levels and population structuring [16]. Avoidance from extreme use of stallion lines and/or mare families has special importance to maintain genetic variance and also protect stallion lines as well as mare families during gene preservation [17]. Previous studies of the CPH population involved analyses of phenotype [2,18] and genetic analyses [19,20,21,22,23]. A pedigree analysis has not been done so far, since the Studbook was established only two decades ago.

Therefore, the main objective of this study was a complete evaluation of the status and structure of the breed, associated trends and risks—with special emphasis on pedigree—and its phenotype characteristics and genetic diversity.

## 2. Materials and Methods

### 2.1. Pedigree Analysis

The Studbook data of the registered CPH up to 2020 were analyzed. The genealogical information was traced back to the founder animals, some of which were born in 1975. The created database consists of the pedigree information for 28,483 horses. For each animal in the database, information such as the UELN, name of the individual, date of birth, sex, and pedigree information were collected. The reference populations consist of those animals that are potentially contributing to the next generation. In this study, animals with both parents known were chosen as the first reference population (REF 01) and active (living) horses of both sexes were selected as the second reference population (REF 02).

Pedigree analysis was conducted using the ENDOG v4.8 software [16]. Several analyses were carried out to evaluate the pedigree and population parameters. Pedigree completeness was observed through the mean number of complete generations, the mean number of maximum generations, and the mean number of equivalent generations [24].

Parameters related to inbreeding (mean inbreeding coefficient (*F*), mean average relatedness coefficient (*AR*), effective population size (*Ne*)), probability of gene origin (effective number of founders (*f_e_*), effective number of ancestors (*f_a_*), and effective number of founder herds (*f_h_*)), and genetic contributions of the founders and ancestors (genetic conservation index (*GCI*) and generation interval (*GI*)) were assessed. The ratio between the effective number of founders (*f_e_*) and effective number ancestors (*f_a_*) as an indicator of a population bottleneck were calculated according to Boichard et al. [24]. Generation intervals (*GI*) were calculated as the average age of the parents at the birth of the progeny subsequently used for reproduction, as well as the average age of the parents of all foals [13,25], and were calculated for the following four paths: stallion—daughter/son, and mare—daughter/son. The four pathways were compared pairwise using paired samples *t*-tests. The average relatedness coefficients and individual inbreeding coefficients were calculated based on Gutiérrez and Goyache [16].

### 2.2. Phenotype Analysis

To see the conformation of CPH, the basic body measurements were taken from 292 licensed stallions and in 255 mares licensed at the first selection (the age of the stallions and mares was between three and four years). The current breeding program [26] prescribes the licensing procedure and three reference body measurements must be taken: height at withers (WH, with a Lydtin rod), chest circumference (CC), and cannon bone circumference (CB, both with a tape measure). These basic phenotype measurements are necessary to gain a more complete insight into the situation and trends in the population of the CPH. All measurements were taken on the left side of the horse while standing on a horizontal surface in standard position.

### 2.3. Genetic Analysis

Hair root samples were collected from the mane of 292 stallions recorded in the CPH Studbook. These stallions originate from all 29 active sire-lines and the number of stallions per line varied from 1 to 57. Isolation of DNA and genotyping were performed in an authorized laboratory. A total of fifteen microsatellite markers (AHT4, AHT5, ASB17, ASB2, ASB23, HMS2, HMS3, HMS6, HMS7, HTG4, HTG6, HTG7, HTG10, and VHL20) were used, as recommended by the International Society for Animal Genetics (ISAG) for paternity confirmation.

Allele frequencies, average (*N_A_*) and effective (*N_AE_*) number of alleles, unbiased estimates for observed (*H_O_*) and expected (*H_E_*) heterozygosity, Shannon’s Information index (*I*), and migration rate (*Nm*) were calculated using GenAlEx version 6.5 [27,28]. The same program was used to test the level of genetic differentiation between sire-lines. The relative Shannon’s Information index is calculated according to Gáspárdy et al. [29]. The polymorphism information content (*PIC*) of each locus and null allele frequency (*F_(null)_*) were quantified using Cervus software version 3.0.7. [30]. To perform the exact test for Hardy–Weinberg equilibrium (*HWE*) using the Markov chain method (10,000 iterations, 100,000 dememorizations, and 1000 batches) and to test the linkage disequilibrium (*LD*), to determine the extent of distortion from independent segregation of loci, the GenePop 4.3 package was used [31]. The inbreeding coefficient (*F_IS_*) according to Weir and Cockerham [32] was calculated as implemented in the Genepop 4.3 software [31]. Individual observed heterozygosities were regressed on individual inbreeding coefficients using the Pearson correlation coefficient. Genetic relationships among individuals were represented using factorial correspondence analysis (*FCA*) as implemented by Genetix 4.05 [33]. Nei’s standard genetic distance [34] was estimated to obtain a neighbor-joining tree in the program Splittree4 version 4.13.1 [35] to visualize the genetic relationship. Whether the CPH stallions experienced a genetic bottleneck was examined using the Bottleneck 1.2.02 program [36] with the Wilcoxon sign-rank test for heterozygosity excess under the assumption of a two-phase mutation model [37]. The second genetic bottleneck test was confirmed by a mode-shift indicator test based on a qualitative descriptive allele frequency distribution [38].

## 3. Results

### 3.1. Pedigree

The trend in demographic data of the CPH population is shown in Figure 1. The number of registered horses per year grew slowly until the end of the 20th century. After the establishment of the CPH Studbook, the number of registered animals increased. In the last two decades, the number of registered animals was between 1000 and 1500 per year, and it was stable with only some fluctuations.

An analysis of all pedigree entries of CPH revealed 25,263 animals for which both parents were known (REF 01, Table 1). The number of founder animals was 1486 and the effective population size of the founders is 162.56. The active living reference population (REF 02) consisted of 5854 active horses.

The mean number of maximum traced generations in the Studbook was 3.83, and the mean number of complete generations was 1.66. The mean number of the equivalent complete generations was 2.42. Figure 2 shows pedigree completeness of the reference populations (REF 01 and REF 02).

The REF 02 had a completeness value for the first two generations (99.04% and 96.66%), while in the third generation the completeness dropped to 80.76% compared to the REF 01 population (92.59%, 77.92%, and 51.96%). Beyond the seventh parenteral generation the completeness was close to zero.

The estimates of the average age of the parents at the birth of their offspring were generally higher in the maternal pathway (8.10 ± 4.15, mother—daughter; 8.29 ± 4.08, mother—son) compared to the paternal pathway (*p* < 0.01, Table 2), where the lowest pathway of 6.78 ± 2.56 years was found for the father—son relationship. Considering all the pathways, the average age of the parents at the birth of their offspring was equal to 7.63 ± 3.55 years. The average generation interval for CPH were 8.20 ± 3.73 years and ranged from 7.99 ± 3.08 (father—daughter) to 8.37 ± 4.20 (mother—daughter), with the average generation interval being longer in mares than in sires (*p* < 0.01).

The effective number of founders (*f_e_*) and ancestors (*f_a_*) was 138 and 107, respectively (Table 1). The ratio between *f_e_* and *f_a_* is 1.29, indicating the existence of a bottleneck in the populations.

The average inbreeding (*F*) in the total population was 0.85%, ranging from a minimum of 0.11% (1999, beginning of breed consolidation and a larger number of founders) to a maximum of 0.90% (2009) (Table 1). In the last decade, the average inbreeding range was from 0.82 to 0.87, indicating a stabilization of the population and a very good implementation of the breeding program (Figure 3). The mean average relatedness (*AR*) coefficient in the whole population was 1.08%. The effective population size calculated via individual increase in inbreeding is 170.89 and via regression on equivalent generations is 240.12.

The mean value of the genetic conservation index (*GCI*) was 4.46 ± 2.42 and ranged from 0.79 to 16.70. The mean value of the *GCI* in REF 01 was lower compared to the REF 02 population (4.86 ± 2.26 vs. 5.83 ± 2.46). The *GCI* calculated for sex was 4.32 ± 2.45 in females and 4.62 ± 2.38 in males.

The Wright *F* parameters *F_IS_*, *F_ST_*, and *F_IT_* were calculated based on the living population of the CPH (REF 02), which consisted from 29 sire-lines and 152 mare-lines (22 mare-lines with less than ten offspring were not included in the analysis). Mean co-ancestry within sire-lines was 0.056 and the Nei distance between the sire-lines was 0.048. The parameters of the *F* statistic related to the sire-lines were *F_IS_* = −0.049, *F_ST_* = 0.049, and *F_IT_* = 0.002. The mean co-ancestry within mare-lines was 0.135 and the Nei distance between the mare-lines was 0.127. The parameters of the *F* statistic related to the mare-lines were *F_IS_* = −0.144, *F_ST_* = 0.1281, and *F_IT_* = 0.002.

### 3.2. Morphometric Variability

Regarding the basic morphometric measurements, it was found that the average WH of the stallions was 142.79 ± 3.41 cm, while the mares were 3.08 cm lower (139.71 ± 2.95 cm). The average CC of the stallions was 194.28 ± 7.63 cm, while the average CC of the mares was 3.98 cm lower (190.30 ± 7.29 cm). The average CB of the stallions was 22.34 ± 0.99 cm and is 1.40 cm higher than that of the mares (20.94 ± 0.96 cm). The differences in body measurements between the stallions and mares were significant (*p* < 0.01).

### 3.3. Microsatellite Variability and Genetic Diversity Indices

A total of 292 CPH stallions were genotyped and the genetic diversity indices are presented in Table 3.

Thirteen loci (93%) had *PIC* values above the threshold (*p* > 0.5) required for their suitability for genetic studies, ranging from 0.630 (HMS7) to 0.822 (ASB17), while only one locus (HTG6) was reasonably informative (*PIC* = 0.301). The mean value for the *PIC* across all loci was 0.697. The estimated null allele frequency (*F_(null)_*) varied slightly between negative (−0.040 AHT4) and positive (0.024 HTG7) values, which were close to zero and assuming the absence of a null allele. Therefore, all 14 microsatellites were retained for the genetic analysis. However, it is possible that the null allele frequency is underestimated if the sample size is small; therefore, the number of 292 individuals of the CPH is sufficient for accurate estimation of the genetic parameters [39].

The total number of alleles was 111, with a mean allele number per locus of 7.9, ranging from a minimum of 5 (HTG7) to a maximum of 13 alleles (ASB17, Table 3). The average observed heterozygosity (*H_O_*) was 0.730 and expected heterozygosity (*H_E_*) averaged 0.731. All loci examined were in Hardy–Weinberg equilibrium (Table 3). The mean Shannon’s Information index was 1.577 and ranged from 0.719 (HTG6) to 2.00 (VHL20), while the relative Shannon’s Information index fell in the range between 40.17–87.79%, with a mean of 73.78% (Table 3).

The inbreeding coefficient (*F_IS_*) was calculated based on *H_E_* versus *H_O_* and ranged from −0.076 (AHT4) to 0.046 (HTG7). The mean inbreeding value of −0.015 was not significant and did not confirm the evidence of homozygote excess in CPH stallions. Two loci of AHT4 showed a significant (*p* < 0.05) excess in heterozygotes of 7.6% and 5.4% (Table 3). The Pearson correlation test performed for CPH showed a negative, non-significant correlation between the inbreeding coefficients and heterozygosity.

Several analyses were carried out to detect possible inter-line structuring, none of which revealed any evidence of clustering (Figure 4). Although they are representatives of different sire-lines, an inter-line subdivision is possible due to the short pedigree (the Studbook was opened very late, in 1998).

An *FCA* scatter plot was constructed including all animals and loci to summarize the individual relationships, with two principal factors explaining 5.97% of the total variation (not shown). With the exception of five individuals that were dislocated from the rest of the population, the other individuals were evenly scattered along the two dimensions of the *FCA* and the lines overlapped with no indication of grouping. The results show no clear separation and indicate that the CPH stallions studied are genetically heterogeneous.

To gain more insight into the relationships within lines from the 29 sire-lines, we left in the analysis those that were represented by five or more genotyped individuals. Thus, of the 29 sire-lines, 13 remained, with a total of 252 individuals (range per line was from 5 to 57). Genetic differentiation among the 13 most representative sire-lines was negligible, with an average *F_ST_* value of 0.02. The high number of migrants (*Nm*) of 12.4 influenced such a low differentiation. In addition, the low genetic variation due to inter-line differences (2%) is in agreement with the above, while differences within individuals accounted for the remaining 98% of the variation. The overall pairwise relatedness was low (−0.008) among the 252 individuals from the 13 sire-lines. This indicates that members were either unrelated or very distantly related. Pairwise relatedness values above 0.50 (expected of a parent/full sibling) were detected once, values from 0.25 to 0.50 (grandparent level/half-sibling) were rare (0.25%), and pairwise relatedness with values below 0.25 was in the majority of cases (99.75%).

The Wilcoxon test gave a significant result (*p* = 0.0158), suggesting that the CPH may have passed through a recent bottleneck. The second method, the mode-shift indicator test, detected normal L-shaped curves, in which the alleles with low frequencies were the most numerous and have a normal distribution [37].

## 4. Discussion

The breeding consolidation activities of the CPH were more intensively pursued in the late 1990s, after an inventory of the remaining population was carried out and the breed standards were defined. For this reason, between 1995 and 1999, 101 to 195 individuals were registered annually in the Studbook, while in 2000 more than 1000 individuals were registered, and this trend was maintained over the next two decades (Figure 1). Furthermore, the annual entry of new individuals into the register is a result of the reproductive vitality due to the economic use of the breed (production of horse meat in the traditional extensive system, see Appendix A).

The shallow depth, i.e., number of generations of the Croatian Posavina horse pedigree (mean number of equivalent generations, 2.42), is a result of the lack of systematic selection for this breed in the past century. Local horse breeds in Croatia were not selectively built up in the 20th century, and at the end of the 20th century, the Studbook was established, which is the basis for the selection and preservation of the breed. Horse breeds that have a long breeding tradition have a bigger pedigree depth and pedigree completeness. For example, Zechner et al. [7] for the Lipizzan horse determined 15.2 equivalent generations while Pjontek et al. [11] for the Lipizzan horse in Slovakia determined 10.25 equivalent generations. The Hucul horse population is a regional horse breed with different mean numbers of equivalent generations (7.10, 6.83, and 9.10) in Slovakia, Czech Republic, and Hungary, respectively [11,14,40]. Turkish Arabian horses have a deeper pedigree (7.8) compared to Spanish Arabian horses (5.7) [13,41]. The American Shire horse also has a favorable pedigree depth (8.22) [42], similar to the Andalusian horse (8.26) [8], but less than the Austrian Noriker horse (12.3) [10]. Although the average number of generations that can be traced in the pedigrees of CPH is not large, in the program of breed conservation, the pedigree records are of great importance. Giontella et al. [15] suggest that knowledge of the relationships among horses is essential for genetic management of breeds and represents one of the principal tools employed to optimize their conservation strategies.

Generation intervals are important factors of population management measures [11]. The GI in the CPH population was 8.20 years, which is consistent with the observation that horse breeds are mated earlier for production purposes, and therefore the GI is shorter [13]. This result is comparable with the GI of other coldblooded type of horses, e.g., Czech coldblooded horses (8.53 years for the Silesian Norik, 8.56 years for the Czech-Moravian Belgian, and 8.88 years for Norics [4]) and the Austrian Noriker (7.9 years) [10]. Warmblood horse breeds usually have a longer *GI*: 10.1 years for the Andalusian [8], 10.65 years for the Maremmano horse [43], and 12.4 years for Turkish Arab horse [13]. In the case of CPH, a shorter *GI* is a consequence of the breed’s economic use for meat production (foal meat), with early initiation into reproduction (three to four years) and regular fertility. Reproductive vitality of the population is desirable in local endangered breed conservation programs.

The number of founders in the CPH was 1486 while the effective number of founders was 138 or 9.29%. A lower *f_e_* value was observed in the Slovak, Polish, and Hungarian Hucul horse (26, 40, and 22, respectively) [11,14,44]; Andalusian horse (39.6) [8]; Spanish and Turkish Arabian horse (38.6 and 40.0) [13,41]; Maremmano horse (74) [43]; Lipizzaner (94) [11]; American Shire horse (104.5) [42]; and Austrian Noriker (117.2) [10]. The effective number of ancestors was 107 or 7.76% of the total ancestors (1435). The rate of the effective number of founders compared to the effective number of ancestors can be used to determine the bottleneck in the population, and if the ratio is 1, the population is stable. A larger ratio reflects a more severe bottleneck effect [24]. The bottleneck ratio (*f_e_*/*f_a_*) in the CPH population was 1.29, which is more favorable compared to the Austrian Noriker (4.00) [10]; American Shire (3.65) [42]; Silesian Norik, Noriker horse, and Czech-Moravian Belgian horse (3.11, 2.33, and 1.86, respectively) [45]; and Maremmano horse (2.4–2.5) [43]. The observed *f_e_*/*f_a_* ratio in the Slovak, Polish, and Hungarian Hucul horse was 1.6, 2.5, and 1.4, respectively [11,14,44]. The high *f_e_*/*f_a_* ratio reveals a disproportionate use of some breeding animals, presumably stallions, resulting in a loss of genetic diversity compared to that expected under random mating conditions [42]. Although the results of the current study indicate a bottleneck, it was not large enough to have an effect on the population.

The level and trend of the inbreeding coefficient in the population are crucial for maintaining the genetic diversity of the breed. The inbreeding coefficient in the CPH population is 0.85% and is lower than in some other coldblooded horse populations, such as the Norik, Silesian Norik, and Czech-Moravian Belgian, with 2.00%, 4.60%, and 4.00% [45], and the Austrian Noriker with 5.00% [10]. Usually, higher *F* values are reported in the case of closed Studbooks and sport-oriented horses, e.g., 8.5% in Andalusian horses [8] or 4.02 in Lipizzan horses [11]. The observed low *F* is followed by a low *AR* (1.08), indicating a reduced representation of each individual in the whole population. This suggests that the CPH population is large in population size, that the stallions used for reproduction have a limited number of services within their own herd (controlled by the breeder), and that the interchange with other herds is frequent so there will be no difficulties trying to avoid mattings between relatives. Horses with the lowest *AR*, used as stallions and mares, can reduce inbreeding and balance the gene contributions of the founders in the population and thus the genetic variability [12]. High values of the inbreeding coefficient, average relationship coefficient, individual increase in inbreeding, and low values of effective population size indicate a loss of genetic variability and possible phenotypic expression of genetic defects.

The *GCI* in the CPH population (original dataset) was relatively low (4.46) and slightly higher (5.83) in the living REF 02 population. In general, the trend in *GCI* in the CPH population increased over time. Giontella et al. [43] reported higher values for a *GCI* of 5.55 in the local Italian Maremmano horse breed. The *GCI* value of the actual breeding population of the Hucul horse in Hungary was 15.1, increasing from 8.9 (population born before 1990) to 15.3 (population born between 2010 and 2019) [14]. In general, animals with higher *GCI* values exhibit a greater balance in the number of founders and it is assumed that it contains the genes transmitted by the founders. From this point of view, animals with a higher *GCI* value are crucial for breed conservation and the genetic variability they possess.

It is important to emphasize that if the effective population size is favorable (*Ne* = 438.71), all 29 sire-lines and 174 mare-lines should be maintained because, with careful mating management, they offer the possibility of sustainability of the genetic diversity. The size and diversity of the sire-lines and mare-lines should be balanced.

The basic morphometric measurements of the CPH from this study compared to previous study of Ivanković and Caput [2] remain largely unchanged. Stallions and mares in this research have a smaller WH of 0.74 cm and 1.14 cm. Moreover, the CC of the stallions were lower at 5.71 cm, while the circumference of the CB increased by 0.44 cm for the same breed. On the other hand, the CC of the mares were similar (190.37 vs. 190.18 cm) as well as the circumference of the CB (20.95 vs. 20.94 cm). Compared with the body measurements of the stallions and mares of half a century ago [46], there is an increase in height at withers (+4.75 cm in stallions, +3.68 cm in mares), chest circumference (+40.62 cm in stallions, +30.45 cm in mares), and tibia circumference (+3.82 cm in stallions, +3.30 cm in mares). The increase in body measurements is reflected by the larger body frame of today’s CPH population, as a result of the selective and limited introduction of coldblooded breeds and the better feeding.

These morphometric measurements are in the group of “A—desirable” breeding standards; therefore, the studied stallions are a good example of representatively chosen lines for further breeding according to the Breeding Program for Croatian Posavina Horse [26]. To be more precise, the Breeding Program for Croatian Posavina Horse has three classes for measured traits defined as “A—desirable”, “B—acceptable”, and “C—undesirable”. In the initial phase of the implementation of the program for the protection (and selection) of locally endangered breeds (i.e., population inventory, establishment of the Studbook), greater phenotypic variability is acceptable due to a more comprehensive coverage of the remaining population [47,48]. This approach was applied to the remnant CPH population (approximately 1000 individuals) at the end of the 20th century, followed by stabilization of the census size and economic affirmation of the breed, and more recently by forming sire-lines. Significant differences (*p* < 0.01) in body measurements between stallions and mares were expected due to sexual dimorphism.

A characteristic of the CPH that is particularly valued by breeders is its compactness and adaptability to the environment due to a smaller body conformation compared to coldblooded horse breeds. The experience of breeders is that the compact conformation accommodates the lower maintenance requirement of horses during feeding (compared to coldblooded horse breeds), but also leads to a favorable conformation of the carcass and the ratio of tissues in the carcass. The CPH is phenotypically different from the other two autochthonous (local) Croatian horse breeds (Croatian Coldblood horse and Murinsulaner horse). For example, the CPH is smaller in withers height (−9.0 cm for stallions and −8.8 cm for mares) compared to the Croatian Coldblood horse [2]. In addition to body size frame, the CPH has a more refined (lighter) head, a thinner cannon bone (smaller circumference), and a lower food need.

Phenotypic measures, as well as favorable interior qualities (i.e., calm temperament, good character, and strong willingness to work) are often neglected in breeding conservation management, although they are a fundamental and immediate tool for the implementation of the desired selection measures, especially by breeders. Indeed, the breeders do not only care for of the genetic purity and herd structure, but they also improve the economic potential of animals through, e.g., meat production. According to a principle of the conservation of genetic resources, “we use it or we lose it”, breeding consolidation should include and weigh all sources of information to find the optimal model of “maximum conservation of genetic potential with maximum income from local breeds”. Maintaining the good interior qualities of the horse that makes it suitable for work in agriculture and agrotourism as well as for other uses (meat production, hippotherapy, recreational riding, traditional carriage driving, milk production, etc.) is another approach to CPH conservation.

The high *PIC* values of the 13 microsatellite loci (93%) used in the study showed that the markers were suitable for genetic evaluation of CPH. The reason for the lower polymorphism of the HTG6 marker was the high frequency of the allele O (*f* = 0.83) and the high percentage of homozygous individuals (68%) of the total genotyped stallions. Similar results were obtained by Vostrá Vydrová et al. [49] for three native draught horse breeds, who reported that 93% of the markers were highly informative with an overall *PIC* value of 0.674.

In the present study, the analyzed sire-lines of the CPH captured a high amount of genetic diversity, as indicated by the high number of alleles, high gene diversity, and low inbreeding coefficient. When microsatellite loci show large differences in allele number, their effectiveness can be represented by *I_rel(%)_* [29], which was 73.78% in this study, showing considerable variation within CPH. The measure of allele diversity is important for many aspects of genetic variability and can vary greatly between breeds. This measure was high in CPH (7.9) compared to reports from several draught horse breeds: five breeds from Germany (*N_A_* = 5.5–6.33) [50]; four draught horse breeds from the USA, Canada, Chile, and France (*N_A_* = 5.13–6.47) [51]; and two Czech breeds (7.15 and 7.3) [49]. The genetic diversity indices (*N_A_*, *H_O_*, and *H_E_*) of the CPH reported 10 years ago [22] were 5.9, 0.710, and 0.724, respectively, and although the *N_A_* was lower, *H_O_* and *H_E_* are close to the results of this study, indicating preserved genetic diversity of the CPH stallions in recent times. Furthermore, the CPH’s genetic diversity for the common loci was higher compared to five draught horse breeds from Germany (*H_E_* < 0.705) [50] and two native Czech draught horse breeds (*H_E_* < 0.680), but similar to the Noriker (*H_E_* = 0.714) [49]. On the contrary, Juras et al. [52] reported higher values for *H_O_* and *H_E_* in Lithuanian Heavy Draught horses (*H_O_* = 0.760, *H_E_* = 0.722), but a lower total (77) and mean number of alleles (6.4) using 16 microsatellites. The breeding range of the CPH has always been over a large geographical area of Posavina (see Appendix A), and the exchange of horses (stallions, mares, and foals) by breeders has been free and without restrictions (they only take care not to inbreed). Moreover, its ancient origin implicates part of lineages from warmblood and coldblood horses such as the Arabian, Lipizzan, Nonius, and Belgian horse breeds [18], indicating a broad initial base of genetic diversity. As expected, this high genetic input from other breeds, observed by the high number of alleles, contributes no trace of erosion of genetic variability or inbreeding.

The *F_IS_* value for the CPH was less than zero, indicating heterozygote excess, which did not cause a deviation from *HWE* (Table 3). In previous studies, the average *F_IS_* value for coldblood horses was reported very differently (from −0.093 to 0.021) [22,51,52]. Similar results to this study were obtained by Vostrá-Vydrová et al. [49] for three native draught horse breeds, where the inbreeding coefficient was as follows: Silesian Noriker (−0.014), Noriker (−0.005), and Czech Moravian Belgian (−0.002). The large gene pool and the absence of a founder effect undoubtedly contributed to the heterozygote excess, without indication of inbreeding within the examined sire-lines. Kovač et al. [18] in conversation with breeders revealed that in the past there was a certain administrative ban on mating Posavina mares with Posavina stallions. Conversely, it was the duty of farmers to mate Posavina mares with imported stallions of heavier breeds, imposed by state authorities. This uncontrolled crossbreeding helped to maintain a low level of inbreeding and high level of diversity, but also resulted in slight changes in some of the original traits, fertility and weight, conformation, adaptability, and resistance of the CPH (see Appendix A). In addition, such introgression of foreign genes could be the reason for the observed low level of inbreeding [53]. The *FCA* plot (data not shown) further supports this fact showing that only 6% of the total genetic variation is explained by the first two axes, with no differentiation between individuals according to sire-lines. Inter-line differentiation was particularly low (2%) compared to the values reported by Vostrý et al. [54] for the Old Kladruber horse sire lines (7%). When analyzing the relationship among the most representative lines, no genetic (sub)division or inbreeding was detected (*F_IS(13lines_*_)_ = −0.093). This is supported by the absence of inbreeding in the stallion-lines according to the pedigree analysis (*F_IS_* = −0.049). Within mare-lines there were a high excess of heterozygotes (14.4%), a high Nei’s genetic distance, and differentiation between lines, as a result of the numerous lines present and the lower number of offspring from the maternal line. Third- and higher-order individuals related to each other indicate that the CPH population is largely composed of multi-generational family units, due to the likely broad genetic base.

In this study, due to the high number of alleles (*N_A_* = 7.9) and high genetic variability (*H_E_* = 0.731) within the CPH population, strong evidence of a recent population bottleneck was not confirmed. Although the CPH population experienced a demographic decline in population size [18], Ivanković and Caput [2] and Galov et al. [23] reported no evidence of a genetic signature of a bottleneck.

Population heterogeneity of the CPH is desirable because it preserves the necessary genetic variability and breeding survival rates in an uncertain future. However, from the perspective of breeding and selection, it is necessary for the lines to differ to some extent while maintaining breed characteristics. Even in the past, two types of CPH horses were mentioned [55]. However, this distinction between the lighter and heavier CPH types was lost when the number of horses was significantly reduced in the 1980s. The remaining individuals of the CPH served as a nucleus for breed revitalization. In the future (after the breed is fully consolidated), the CPH may retain its current characteristics of a smaller, coldblooded type horse (which is consistent with the current breeding program), or be profiled toward a lighter type that can be used for recreation, performances at shows and traditional events, recreation, etc. The results of the study will have a practical application in the selection of the CPH population and the development of an annual mating plan, especially in terms of confirmation of the less represented stallion lines and regular monitoring of their population parameters (*F*, *GI*, etc.). The results of the study show that the preserved structure of the population and its genetic diversity justify the closure of the Studbook. Profiling of a lighter riding horse within the CPH population is justified after the interest of the breeders has been expressed and the CPH no longer has an endangered status. In addition, this decision should take into account the indicators of their phenotypic and genetic structure and evaluate the economic impact after the change in breeding objectives.

## 5. Conclusions

A complex study of the pedigree, phenotype, and genetic structure of the breed reveals the need for a comprehensive approach to dynamic population monitoring of the Croatia Posavina horse. Namely, breeders of the Croatian Posavina Horse monitor the horse’s phenotype and primarily select the best individuals for future mattings and, based on that, for breeding, while the breeding association and the scientific institutions involved provide additional guidelines based on the pedigree and genetic analysis of the population. The results of the present study show that the Croatian Posavina horse has valuable amount of genetic variability with no signs of inbreeding due to the large gene pool. There was no evidence of clustering and phenotypic or genetic differentiation in the population of the Croatian Posavina horse. We can also conclude that the Croatian Posavina horse has maintained its phenotypic recognizability, compactness, and has a favorable number of lines of stallions and mares. This breed, with the appropriate breeding management (and economic reaffirmation), can maintain the existing genetic variability and, with the size of the population and structure, leave the category of an endangered breed. With these results, comprehensive information will be obtained for further breeding management and conservation of the breed.

## Figures and Tables

**Figure 1 animals-11-02130-f001:**
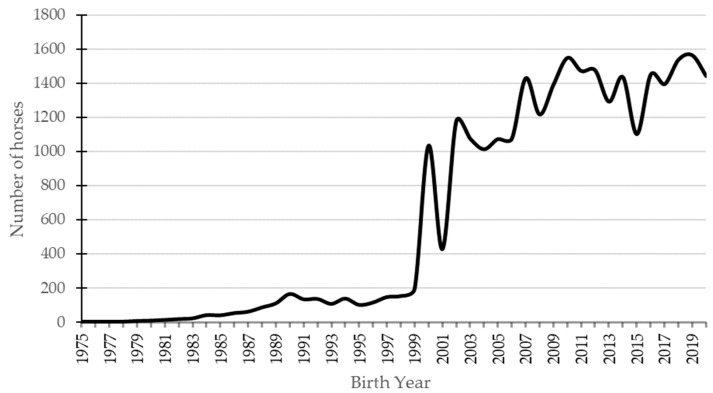
The number of horses registered in the CPH Studbook by birth year.

**Figure 2 animals-11-02130-f002:**
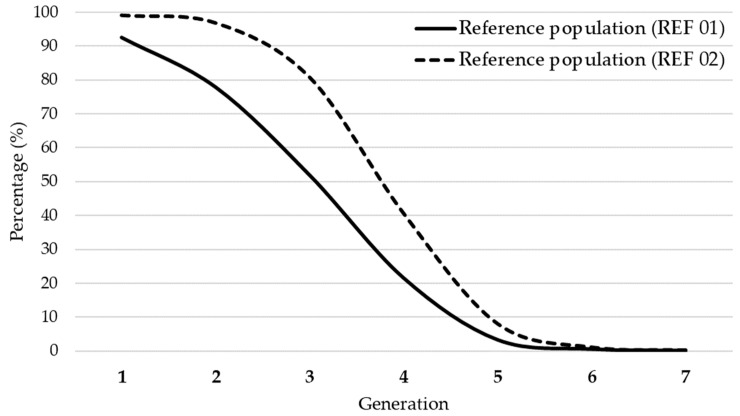
Average percent of pedigree completeness for the REF 01 and REF 02 reference populations.

**Figure 3 animals-11-02130-f003:**
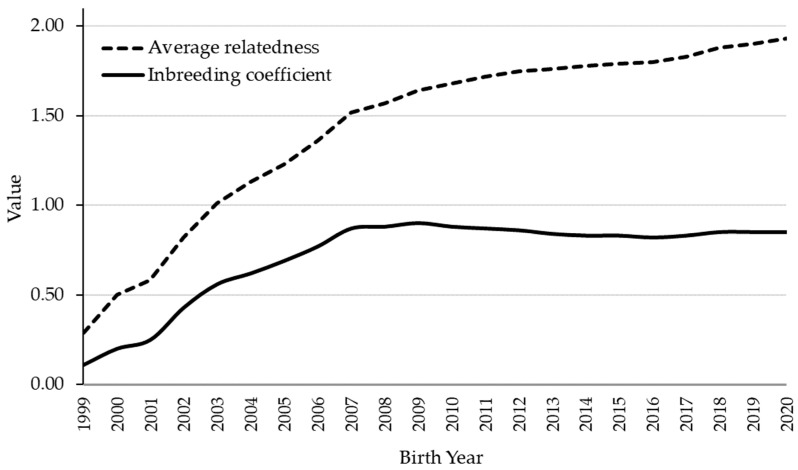
Average level of inbreeding coefficient and average relatedness in the Croatian Posavina horse.

**Figure 4 animals-11-02130-f004:**
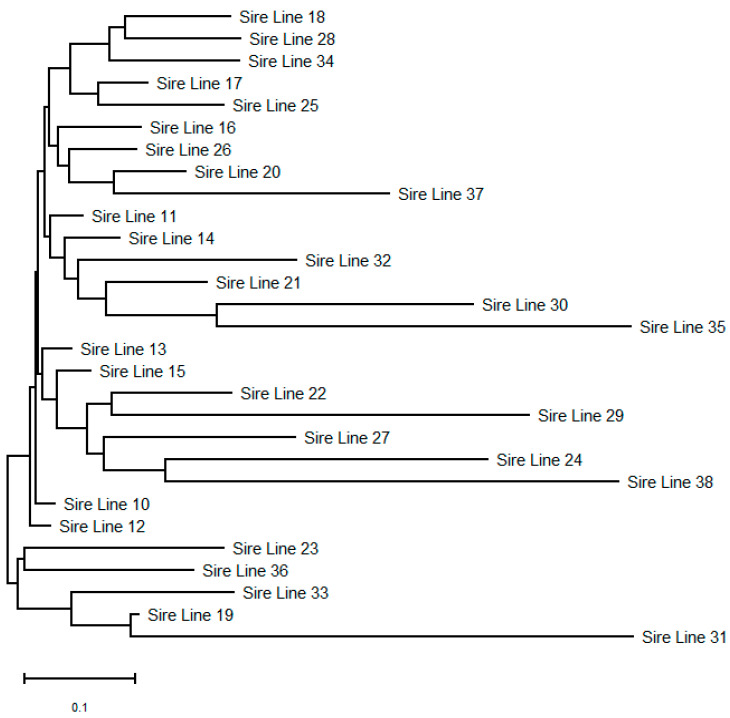
Unrooted tree showing relationships among 29 sire-lines of the Croatian Posavina horse.

**Table 1 animals-11-02130-t001:** Investigated CPH population size and genetic diversity indices.

Item	Total
Original dataset (TP; total population)	28,483
Reference population 1 (REF 01; animal with both parents known)	25,263
Reference population 2 (REF 02; live animals)	5854
Number of founders contributing to reference (REF 01) population	1486
Effective number of founders (*f_e_*)	138
Effective population size of founders	162.56
Number of ancestors contributing to reference (REF 01) population	1435
Effective number of ancestors (*f_a_*)	107
Number of ancestors explaining 50% of genetic variability	46
Effective number of founders/effective number of ancestors (*f_e_*/*f_a_*)	1.29
Number of founder herds in reference population (*f_h_*)	466
Effective number of founder herds for the reference population	61.90
Effective population size (*Ne*)	438.71
Effective population size reference REF 02 population (computed via individual increase in inbreeding)	170.89
Mean inbreeding (*F*)	0.85
Mean Average Relatedness coefficient (*AR*)	1.08
Genetic Conservation Index (*GCI*)	4.46
Mean number of maximum generations	3.83
Mean number of equivalent generations	2.42
Mean number of complete generations	1.66

**Table 2 animals-11-02130-t002:** Average age of parents at the birth of their offspring and generation interval (years) for the Croatian Posavina horse (REF 01).

Pathway	*n*	Mean ± SD	SE
Average age			
Father—son	445	6.78 ± 2.565 ^A^	0.122
Father—daughter	4734	7.17 ± 2.734 ^B^	0.034
Mother—son	442	8.29 ± 4.079 ^C^	0.194
Mother—daughter	4934	8.10 ± 4.146 ^C^	0.059
Overall	10,555	7.63 ± 3.550	0.035
Generation interval			
Father—son	12,074	8.07 ± 3.090 ^a,A^	0.028
Father—daughter	13,521	7.99 ± 3.083 ^b,A^	0.026
Mother—son	12,896	8.35 ± 4.273 ^B^	0.038
Mother—daughter	14,254	8.37 ± 4.197 ^B^	0.035
Overall	52,745	8.20 ± 3.725	0.016

SD, standard deviation; SE, standard error of the mean; different superscript capital (*p* < 0.01) or small (*p* < 0.05) letters show significant differences.

**Table 3 animals-11-02130-t003:** Parameters of the genetic diversity for each of the 14 microsatellite markers in stallions of the Croatia Posavina horse (*n* = 292).

Locus	*N_A_*	*H_o_*	*H_E_*	*PIC*	*I*	*I_rel(%)_*	*HWE*	*F_(null)_*	*F_IS_*
AHT4	8	0.767	0.712	0.670	1.494	71.85	ns	−0.040	−0.076 *
AHT5	7	0.774	0.768	0.740	1.655	85.09	ns	−0.009	−0.006
ASB17	13	0.884	0.837	0.822	2.096	81.75	ns	−0.029	−0.054 *
ASB2	8	0.818	0.791	0.760	1.681	80.84	ns	−0.019	−0.033
ASB23	9	0.795	0.777	0.744	1.727	78.56	ns	−0.014	−0.021
HMS2	9	0.736	0.764	0.735	1.706	77.61	ns	0.022	0.038
HMS3	6	0.733	0.731	0.688	1.483	82.86	ns	−0.002	−0.001
HMS6	7	0.753	0.722	0.675	1.426	73.32	ns	−0.022	−0.042
HMS7	7	0.668	0.678	0.630	1.352	69.51	ns	0.011	0.016
HTG10	10	0.836	0.801	0.774	1.831	79.52	ns	−0.024	−0.042
HTG4	6	0.750	0.737	0.697	1.490	83.25	ns	−0.010	−0.015
HTG6	6	0.318	0.311	0.301	0.719	40.17	ns	−0.003	−0.022
HTG7	5	0.712	0.745	0.699	1.413	87.79	ns	0.024	0.046
VHL20	10	0.842	0.839	0.820	2.000	86.86	ns	−0.003	−0.002
Average	7.9	0.730	0.731	0.697	1.577	73.78			−0.015

Number of alleles (*N_A_*); observed (*H_O_*) and expected (*HE*) heterozygosities; Shannon’s Information index (*I*); relative Shannon’s Information index (*I_rel(%)_*); polymorphism information content (*PIC*); deviation form Hardy–Weinberg equilibrium (*HWE*); frequency of null alleles *F_(null)_*; fixation index (*F_IS_*); ns—non significant; * *p* < 0.05.

## Data Availability

The data presented in this study are available on request from the corresponding author. The data are not publicly available to preserve privacy of the data.

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
