# Peer review of "Evaluation of the Conservation Status of the Croatian Posavina Horse Breed Based on Pedigree and Microsatellite Data"

_animals, 2021, doi:10.3390/ani11072130_

Round 1
Reviewer 1 Report
Dear Sir,
The manuscript entitled “Complex Evaluation of Population Structure in Developing Strategies for Preservation Local Horse Breeds: an Example of the Croatian Posavina Horse” evaluated the genetic diversity in the Croatian Posavina Horse using pedigree information and microsatellite data. It gives a nice overview of the origins and population numbers of the breed. However, the title does not really fit with the main aspects of the manuscript, so I would change the title to “Evaluation of the conservation status of the Croatian Posavina Horse breed based on pedigree and microsatellite data”.
I still have some questions to be addressed, notably on the interpretation of the necessity of maintaining stallion lines, if they cannot be distinguished either phenotypically or genotypically. It is also regrettable that no direct comparison was made to other breeds, i.e. there is no justification that this breed is distinct from other European breeds and should be conserved. In fact, the tested individuals seem less inbred because the admixture from other breeds is quite large. This fact should be discussed, even just to say that a comparison with other breeds could not be done for lack of available data. In addition, the genetic analysis was performed on microsatellites, while most modern studies use a SNP array of medium or high density. This should also be mentioned as a limitation to this study in the discussion.
The English was quite good, but the verb tenses were not used consistently. Another check by a native speaker is necessary before publication.
I would recommend to publish this article after thorough revisions of the text.
Detailed comments:
Title:
I would suggest: “Evaluation of the conservation status of the Croatian Posavina Horse breed based on pedigree and microsatellite data”.
Abstract
Page 1, line 26: the aim of “this project” or “this study”, “was to analyze”…
Page 1, line 28-29: replace “and” by “±”: 8.20 ± 1.66 years
Page 1, line 35: there “was no evidence of inbreeding”….
Introduction
Page 2, line 47-48: I do not understand this sentence, please rephrase
Page 2, line 50: parentheses missing. ) and [
Page 2, line 58: you do not need “through their importance” in this sentence
Page 2, line 61: “was at the intersection of the Habsburg and Ottoman Empires”
Page 2, line 65: do you mean “needs” instead of “options”?
Page 2, line 68: “has retained” or “retained”
Page 2, line 70: “carcass composition” instead of “conformation”
Page 2, line 71: “in the 20th century”
Page 2, line 72: in the nineteen nineties
Page 2, line 73: “the CPH population declined to about 450 breeding individuals”
Page 2, line 81: I do not understand your statement in parentheses. Please rephrase.
Page 2, line 82: preserving “a breed” or “breeds”
Page 3, line 97: “was the complete…”
Material and methods:
Page 3, line 105-108: please rephrase, the English is not sound here
Page 3, line 115-118: it is not clear what was calculated to assess what. Please rephrase.
Page 3, line 118: word missing?
Page 3, line 123: “The average”,
Page 3, line 124: “coefficients were calculated based on Gutiérrez….”
Page 3, line 136: you do not need to write “mane region”, “from the mane” is enough.
Results:
Page 4, line 165-172: please be careful to separate discussion from results. This should be in the discussion.
Page 4, line 176: replace “where” by “for which”
Page 4, line 177-179: past-tense please. “were” instead of “are”, “was” instead of “is”, “consisted” instead of “consists”
Page 5, line 187-190: please check the English here. There are articles and commas missing. “the seventh” instead of the “seven”.
Page 6: line 202-205: part of interpretation. Should be in the discussion.
Page 7, line 224-225: past tense, “was” and “were”. “ten offspring” instead of “teen”.
Page 7, line 236-237: part of interpretation, should be in the discussion
Page 7, line 240: this is debatable. It might also mean that the sire lines cannot be distinguished from one another phenotypically. Please elaborate in the discussion. In fact, your results also show that the sire lines cannot be distinguished genetically further down in your results (page 9).
Discussion
Page 10, line 325: please reword, I do not understand what you are trying to say.
Page 10, line 331: “e.g. 10.1 years for Andalusians, … for Maremmano horses … for Turkish Arabian horses-
Page 10, line 337: “A lower fe value was observed”
Page 10, line 340: “and Austrian Noriker”.
Page 11, line 355: “closed Studbooks”. I am not convinced Andalusians and Lipizzaner are the best example for sport horse breeds. You could use values for German or French sport horses.
Page 11, line 367-369: past tense: “was” instead of “is”, “increased” instead of “increases”
Page 11, line 372: parentheses problem ) and [
Page 11, line 372-375: I do not understand this statement. Please rephrase and elaborate.
Page 11, line 381-384: please check the English here.
Page 11, line 385: past tense “were” instead of “are”.
Page 11, line 387-389: I do not understand how you come to this conclusion. Please rephrase and elaborate
Page 12, line 400: “conformation” instead of “confirmation”
Page 12, line 404: “care for” instead of “takes care”
Page 12, line 421: four draught horse breeds from where?
Page 12, line 427 and line 445: “conversely” instead of “contrary”
Page 12, line 431: I do not understand what you want to say here. Is the origin of the horse mixed because breeding did not used to be controlled, or why do you write of animal migration?
Page 13, line 464: “from the perspective of breeding and selection,” …
Page 13, line 468: “towards a lighter type horse”
Page 13, line 469: replace “manifestation” by “shows”
Author Response
Dear Reviewer,
Thank you for your time and effort in providing valuable feedback on the manuscript and for your insightful comments. We have seriously considered all comments and suggestions. We have made changes to address most of the suggestions and have highlighted the changes in the manuscript.
We have noted a point-by-point response to the comments and concerns.
We hope that we have succeeded in meeting all your requirements.
Yours sincerely,
The authors of the manuscript.

Reviewer 2 Report
In my opinion the paper by Ivankovic et al. is interesting and generally worth publishing. However, I have several questions and comments which should help in better presentation of evaluated work’s outcomes.
General comments:
- Some more information about the CPH breed are needed.
- Is the registry still open for introgression of other coldblooded horse breeds? Based on genetic diversity parameters presented by Authors it is likely.
- Is the CPH breed phenotypically different from other coldblooded horses kept in Croatia?
Taking into account your possible responses for the above questions I have a few more:
- What is the real reason of CPH preservation? I think the phenotypic or genetic uniqueness is not the case here. Is the breed listed on FAO list? If yes, what is the current status of the breed? Lines 80-81: “The actual size of the CPH population is about 3,200 reproductively active individuals (with offspring 5,845 head) but it is still vulnerable” – this is too laconic and do net tell us the true story about reasons of keeping this breed under conservation program.
- Authors indicated that a number of CPHs are currently kept for meat production purposes. So, they are probably selected for body size and carcass composition. How to effectively combine the two goals – selection and genetic diversity preservation? Or maybe there are two distinct populations – one covered by conservation program, and the second one – selected without any deep thinking about genetic diversity parameters? I think this should be discussed shortly in the paper.
- The language of the paper can be improved. There are several unclear sentences. Many of them contain some unnecessary repetitions, e.g. Line 17: relatively…relatively; line 36: diverse…diversity; line 126: measurements were measured; lines: 358-359 CPH population is large in population… etc. Pleased read everything once again and correct.
Specific comments:
- Line 24: change protection program à conservation program
- Line 67: I do not think that introgression of Arabian improved the “power” of heavy draft horse. Maybe the “stamina” would be a better word here?
- Lines 145-146: it is not clear which traits were compared using ANOVA. Please complete.
- Figure 1 caption: The number of horses registered in the CPH Studbook by birth year
- Line 181: do you mean the “equivalent complete generations”?
- Table 1: what is the reason of separation for two sex categories: father-son and father daughter (mother-son and mother-daughter)? Is the sex ratio in CPH not random? Do parents (or breeders) plan to obtain the foal of given sex? ? Please consider the following categories: father-offspring, mother-offspring or give me any convincing explanation of more categories distinguishing. SD and MSE abbreviations should be explained in the Table caption.
- Figure 4: In my opinion if the differences between sire lines are not significant – this figure should be removed. There is an information in the article text, that the population turned out morphologically homogenous, and I think it is enough.
- Table 2 caption sounds weird. It would be better to write: “Investigated CPH population size and genetic diversity indices”
- Lines 231-240: are the morphometric parameters changing in time? It would be interesting to check if the sizes of horses are growing (e.g. for meat production purposes) or maybe some parameters are decreasing (towards “saddle utility”). Is it possible to compare your results with older outcomes of other authors? Looking at the mares from your supplementary file (1950’s vs. modern ones) it is likely that the morphometric parameters increased significantly. This should be discussed in the text too
- Line 263: I am a bit confused here. Do you mean the frequency is “underestimated”? A few sentences before you wrote that the null allele was probably absent.
- Lines: 376-379, again, is the breed really endangered? Or do you expect it will be endangered in the future? Based upon estimated parameters I am pretty sure that CPH population can be even slightly reduced without any drastic negative effects on genetic diversity indices.
- Lines: 485-486 “CPH has maintained its phenotypic recognizability” – it is controversial. I think the breed is phenotypically similar to other native coldblooded horse populations. An again – is the current phenotype identical to those represented by CPH horses living 50-70 years ago?
Author Response

(The authors gave the same response as above.)

Round 2
Reviewer 2 Report
In my opinion the manuscript was significantly improved and can be published in the present form. Good job!
Author Response
Dear Editor, Dear reviewers,
We appreciate the time and effort you have dedicated providing valuable feedback on the manuscript and for all the insightful comments. We took into serious consideration to all comments and suggestions. We have made changes to address most of the suggestions made by the reviewers and have highlighted the changes in the manuscript.
Yours sincerely,
The authors of the manuscript
